# Enhancing Xylanase Production from *Aspergillus tamarii* Kita and Its Application in the Bioconversion of Agro-Industrial Residues into Fermentable Sugars Using Factorial Design

Jose Carlos Santos Salgado [1,2] , Paulo Ricardo Heinen [3], Josana Maria Messias [3], Lummy Maria Oliveira-Monteiro [3] , Mariana Cereia [2], Carem Gledes Vargas Rechia [4], Alexandre Maller [5], Marina Kimiko Kadowaki [5], Richard John Ward [1,3] and Maria de Lourdes Teixeira de Moraes Polizeli [2,3,*]

1 Department of Chemistry, Faculdade de Filosofia, Ciências e Letras de Ribeirão Preto (FFCLRP), University of São Paulo, Ribeirão Preto 14040-900, São Paulo, Brazil; rjward@ffclrp.usp.br (R.J.W.)

2 Department of Biology, Faculdade de Filosofia, Ciências e Letras de Ribeirão Preto (FFCLRP), University of São Paulo, Ribeirão Preto 14040-901, São Paulo, Brazil; macereia@ffclrp.usp.br

3 Department of Biochemistry and Immunology, Faculdade de Medicina de Ribeirão Preto (FMRP), University of São Paulo, Ribeirão Preto 14049-900, São Paulo, Brazil; prheinen@alumni.usp.br (P.R.H.); josana.messias@gmail.com (J.M.M.); lummy@udel.edu (L.M.O.-M.)

4 Department of Biomolecular Sciences, Faculdade de Ciências Farmacêuticas de Ribeirão Preto (FCFRP), University of São Paulo, Ribeirão Preto 14040-903, São Paulo, Brazil; cvrechia@fcfrp.usp.br

5 Center of Medical Sciences and Pharmaceutical, Western Paraná State University, Cascavel 85819-170, Paraná, Brazil; alexandre.maller@unioeste.br (A.M.); marina.kadowaki@unioeste.br (M.K.K.)

* Correspondence: polizeli@ffclrp.usp.br

**Abstract:** The endo-1,4-β-xylanases (EC 3.2.1.8) are the largest group of hydrolytic enzymes that degrade xylan, the major component of hemicelluloses, by catalyzing the hydrolysis of glycosidic bonds β-1,4 in this polymer, releasing xylooligosaccharides of different sizes. Xylanases have considerable potential in producing bread, animal feed, food, beverages, xylitol, and bioethanol. The fungus *Aspergillus tamarii* Kita produced xylanases in Adams' media supplemented with barley bagasse (brewer's spent grains), a by-product from brewery industries. The culture extract exhibited two xylanase activities in the zymogram, identified by mass spectrometry as glycosyl hydrolase (GH) families 10 and 11 (GH 10 and GH 11). The central composite design (CCD) showed excellent predictive capacity for xylanase production (23.083 U mL$^{-1}$). Additionally, other enzyme activities took place during the submerged fermentation. Moreover, enzymatic saccharification based on a mixture design (MD) of three different lignocellulosic residues was helpful in the production of fermentable sugars by the *A. tamarii* Kita crude extract.

**Keywords:** *Aspergillus tamarii* Kita; endo-1,4-β-xylanase; central composite design; mixture design; fermentable sugars; barley bagasse; brewer's spent grains

## 1. Introduction

Large amounts of lignocellulosic materials accumulate in the environment every year through agricultural residues, energy crops, and forestry by-products. In Europe, for example, about 950 million tons of biomass is produced annually, which has the potential for conversion into 300 million tons of oil-equivalent fuel [1].

The decomposition of biomasses by microbial enzymes is of prime interest for using these lignocellulosic residues to produce value-added products and minimize the environmental impacts of their continuous accumulation [2,3]. The biodegradable residues are considered sustainable feedstocks for producing several value-added chemicals [4]. Beer/barley bagasse (BB), also known as brewer's spent grains (*Hordeum vulgare* L.), is a by-product

of the beer industry. This lignocellulosic material constitutes approximately 85% of all by-products generated during beer processing, representing 31% of the original malt weight [5]. BB is rich in hemicellulose (mainly xylose and arabinose units), cellulose (glucose units), and lignin [6,7]. Hemicelluloses are complex mixtures of xylan, xyloglucan, glucomannan, galactoglucomannan, arabinogalactan, and other heteropolymers [8,9]. Xylan is the major component of the hemicelluloses in many plants, comprising between 20 and 35% of the total dry weight in tropical plant biomass [10]. Most xylans occur as complex polysaccharides derived from a homopolymeric backbone of β-1,4-linked D-xylopyranosyl units substituted in varying degrees by D-glucuronopyranosyl, 4-O-methyl-D-glucuronopyranosyl, L-arabinofuranosyl, acetyl, feruloyl, and/or p-coumaroyl side-chain groups. Due to the heterogeneity and complex chemical nature of plant xylans, their complete breakdown into monosaccharides requires the cooperative action of various enzymes [8,9,11]. The xylanolytic system includes the endo-1,4-β-xylanases (EC 3.2.1.8), β-xylosidases (EC 3.2.1.37), α-D-glucuronidases (EC 3.2.1.139), α-L-arabinofuranosidases (EC 3.2.1.55), acetyl-xylan esterases (EC 3.1.1.72), feruloyl esterases (EC 3.1.1.73), and p-coumaric acid esterases (EC 3.1.1.-) [8,9,12].

Currently, the increasing interest in xylanases is related to their importance in different biotechnology processes, such as supplements in poultry, pork, and caprine feed, cellulose pulp bleaching, manufacture of bread and other food, textiles, beverages, and bioethanol production [10,12]. Reducing xylanase production costs is crucial for their economically viable use in industrial applications. Optimizing medium components and growth conditions to increase enzyme production has traditionally been performed by varying one-factor-at-a-time (OFAT) while keeping the other factors constant. This technique is time-consuming and unsuitable for establishing optimum culture conditions, mainly due to the interactions among variables that can also affect the process. Statistical experimental designs, such as the full factorial design (FFD), have been used to replace OFAT to identify the relative variable importance and find optimal culture conditions for maximal enzyme production [13,14].

Lignocellulosic agro-residues are cheaper carbon sources for microorganism growth and have been used as substrates for the production of xylanases [5,15–17]. Filamentous fungi from the genera *Trichoderma* and *Aspergillus* have been studied for many years for their potential in biomass degradation and production of plant cell wall-degrading enzymes (PCWDEs) [18,19]. The genus *Aspergillus* has been used, for example, in the production of enzymes, biodegradation of agro-industrial wastes, and the production of high-value-added products from low-value feedstocks by industrial fermentation processes [20–22]. The saprophytic fungus *Aspergillus tamarii* has been recognized as an efficient producer of proteases [23,24] and xylanases [25–28] and was previously studied by our group [29,30]. These studies have demonstrated that *A. tamarii* is a good producer of xylanases, making it promising for producing other enzymes of biotechnological interest. Furthermore, utilizing BB as a substrate for fungal growth and as a source of sugars contributes to the valorization of this biomass.

In this work, a central composite design (CCD) was used, aiming to increase the production of xylanases by *A. tamarii* Kita using the parameters: BB content (%), volume of liquid medium, and culture time. In addition, a mixture design (MD) containing individual, binary, and ternary formulations of different agro-industrial residues was used to find conditions for obtaining optimal fermentable sugars using these xylanase-rich extracts.

## 2. Material and Methods

### 2.1. Microorganism and Culture Conditions

The fungus *Aspergillus tamarii* Kita (GenBank accession number: KJ995575.1) was maintained on potato dextrose agar (PDA) slants for up to 7 days at 28 °C. After 7 days, 10 mL of sterilized water was added to each slant, and the conidia were scraped off with an inoculation loop. One milliliter ($1.0 \times 10^6$) of conidia was added to 25 mL of culture medium previously autoclaved for 20 min at 121 °C and 1.5 atm. Submerged fermentation (SmF) was carried out in static conditions at 28 °C, using different periods, concentrations of BB, and volumes of Adams' medium (2 g $L^{-1}$ of yeast extract, 1 g $L^{-1}$ of $KH_2PO_4$,

and 0.5 g $L^{-1}$ of $MgSO_4$ $7H_2O$) [31]. After fermentation, the cultures were filtered using Whatman no. 1 filter paper, and the extracellular crude enzyme extracts were dialyzed against distilled water for 12 h before enzymatic assays.

### 2.2. Preparation of Substrates

BB from Martignoni Bier Brewing (Martignoni Bier, Cascavel, Brazil) was used as the substrate for enzymatic production and saccharification assays. In natura, sugarcane bagasse (SCBn) and steam-exploded sugarcane bagasse (SCBp) from sugarcane plants 'Galo Bravo' (Galo Bravo S/A Acucar e Alcool, Ribeirão Preto, Brazil) and Nardini (Nardini Agroindustrial Ltd, Vista Alegre do Alto, Brazil), respectively, are also utilized in combination with BB. These substrates were washed three times with distilled water, oven-dried at 60 °C, and milled in a Wiley-type mill (SL 31; SOLAB, Piracicaba, Brazil) using a 30-mesh sieve.

### 2.3. Optimization of the Submerged Fermentation Process for Xylanase Production

A completely randomized experimental design of multivariate analysis was performed to improve the initial production of xylanases. The conventional OFAT method was initially used to delimit the test range for each of the three variables under study: BB substrate ($X_1$), volume of Adams' medium ($X_2$) [31], and culture time ($X_3$). The levels of the independent variables were defined according to a $2^3$ full-factorial central composite design resulting in 17 experiments (8 factorial runs, 6 axial or star points, and 3 central points) (Table 1). To predict the optimal point, a second-order polynomial function was fitted to correlate the relationship between independent variables and the xylanolytic activity response. Three more central points were added later to validate the final statistical model.

**Table 1.** CCD $2^3$ planning matrix with 3 repeats at the central point for xylanase production.

| Run | Barley Bagasse g ($X_1$) | Adams' Medium mL ($X_2$) | Culture Time h ($X_3$) | Experimental Activity (U mL$^{-1}$) [a] | Predicted Activity (U mL$^{-1}$) [b] | Residue |
|---|---|---|---|---|---|---|
| 1 | −1 (0.400) | −1 (9) | −1 (91) | 17.647 | 16.405 | 1.242 |
| 2 | −1 (0.400) | −1 (9) | 1 (149) | 18.306 | 18.426 | −0.120 |
| 3 | −1 (0.400) | 1 (21) | −1 (91) | 18.776 | 17.448 | 1.328 |
| 4 | −1 (0.400) | 1 (21) | 1 (149) | 19.859 | 19.469 | 0.390 |
| 5 | 1 (0.850) | −1 (9) | −1 (91) | 20.000 | 20.021 | −0.021 |
| 6 | 1 (0.850) | −1 (9) | 1 (149) | 21.365 | 22.042 | −0.677 |
| 7 | 1 (0.850) | 1 (21) | −1 (91) | 21.788 | 21.064 | 0.724 |
| 8 | 1 (0.850) | 1 (21) | 1 (149) | 23.059 | 23.085 | −0.026 |
| 9 | 0 (0.625) | 0 (15) | 0 (120) | 22.634 | 23.011 | −0.377 |
| 10 | 0 (0.625) | 0 (15) | 0 (120) | 23.302 | 23.011 | 0.291 |
| 11 | 0 (0.625) | 0 (15) | 0 (120) | 23.275 | 23.011 | 0.264 |
| 12 | −1.68 (0.250) | 0 (15) | 0 (120) | 15.529 | 16.877 | −1.348 |
| 13 | 1.68 (1.000) | 0 (15) | 0 (120) | 23.294 | 22.952 | 0.342 |
| 14 | 0 (0.625) | −1.68 (5) | 0 (120) | 20.847 | 20.757 | 0.090 |
| 15 | 0 (0.625) | 1.68 (25) | 0 (120) | 21.412 | 22.508 | −1.096 |
| 16 | 0 (0.625) | 0 (15) | −1.68 (72) | 14.965 | 16.571 | −1.606 |
| 17 | 0 (0.625) | 0 (15) | 1.68 (168) | 20.565 | 19.965 | 0.600 |
| 18 | 0 (0.625) | 0 (15) | 0 (120) | 22.824 | 23.011 | −0.187 |
| 19 | 0 (0.625) | 0 (15) | 0 (120) | 23.435 | 23.011 | 0.424 |
| 20 | 0 (0.625) | 0 (15) | 0 (120) | 23.106 | 23.011 | 0.095 |

[a] Each value represents the mean of three enzymatic measures. [b] Values predicted using Statistica 13.0 software.

The significance of each independent variable was checked with Student's *t*-test and *p*-value. The coefficient $R^2$ was used to determine the quality of the fit of the polynomial model, and its statistical significance was evaluated using analysis of variance (ANOVA).

### 2.4. Enzymatic Assays

The endo-β-1,4-xylanase, endo-1,4-β-glucanase (CMCase), exo-1,4-β-glucanase (Cellobiohydrolase/Avicelase), pectinase, and amylase activities were estimated using the 3,5-dinitrosalicylic acid reagent (DNS) (Sigma-Aldrich, St. Louis, MO, USA), as described by Miller [32]. The substrates used for these enzymes were beechwood xylan, carboxymethyl cellulose, microcrys-

talline cellulose, and pectin (from Sigma-Aldrich, St. Louis, MO, USA) and starch (from Reagen, Colombo, Brazil), respectively. The assays were carried out with 50 µL of adequately diluted crude enzyme extract and 50 µL of substrate 1% (*w/v*) in 100 mmol L$^{-1}$ sodium acetate buffer, pH 5.5, at 50 °C for 30 min. The enzymatic assays were stopped with 100 µL of DNS and boiled for 10 min at 100 °C. After cooling, the samples were diluted with 1 mL of distilled water, and 100 µL was used to estimate the reducing sugars at 540 nm in a microplate reader (Molecular Devices, Sunnyvale, CA, USA).

β-glucosidase, β-xylosidase, and α-L-arabinofuranosidase were estimated using *p*-nitrophenyl-β-D-glucopyranoside (*p*NPG), *p*-nitrophenyl-β-D-xylopyranoside (*p*NPX), and *p*-nitrophenyl-α-L-arabinofuranoside (*p*NPA) (Sigma-Aldrich, St. Louis, MO, USA), respectively, as described by Souza et al. [33]. The assays were carried out with 100 µL of crude enzyme extract, 100 µL of buffer (200 mmol L$^{-1}$ of sodium acetate, pH 5.5), and 200 µL of substrate 0.4% (*w/v*). After incubation at 50 °C for 30 min, the enzymatic assays were stopped with 800 µL of sodium tetraborate saturated solution. The release of *p*-nitrophenolate was measured at 410 nm in a microplate reader (Molecular Devices, Sunnyvale, CA, USA). One unit of enzyme activity was defined as the amount of enzyme that catalyzed the release of 1 µmol of reducing sugar or *p*-nitrophenolate per minute under the assay conditions.

### 2.5. Determination of Protein Concentration

The protein content of extracellular crude extracts was determined according to Bradford [34], with bovine serum albumin (BSA) as the standard. The protein values were expressed as mg of protein per mL of solution (mg mL$^{-1}$). The specific activity of the enzymatic extract was expressed as units per mg of protein (U mg$^{-1}$).

### 2.6. Electrophoresis and Zymogram Analysis

Non-denaturing polyacrylamide gel electrophoresis (PAGE) was performed according to the methodology described by Davis [35]. Electrophoresis was performed in 8% polyacrylamide gel (*w/v*) with Tris-glycine buffer pH 8.3 (25 mmol L$^{-1}$ Tris and 192 mmol L$^{-1}$ glycine) under 140 V and 40 mA. The gel was subsequently cut into two parts: one was stained with silver nitrate [36] to visualize the proteins, and the other was used in a zymogram assay to evaluate xylanolytic activity. For the zymogram assay, the gel was incubated in a solution containing 0.5% beechwood xylan (*w/v*) in 50 mmol L$^{-1}$ sodium acetate buffer, pH 5.5 at 55 °C for 30 min. Next, the gel was washed with the same buffer and stained with a Congo red solution (0.2%, *w/v*) for 10 min at room temperature. The gel was destained with 2 mol L$^{-1}$ NaCl solution until the position of the enzymatic activity in the gel was visualized.

### 2.7. Mixture Design (MD)

A simplex centroid MD was used to evaluate the effect and interaction of the mixture of three substrates ((A) BB, (B) SCBn, and (C) SCBp) regarding the release of fermentable sugars by the enzymatic action of *A. tamarii* Kita crude enzyme extract. Each assay comprised 100 mg of residues plus 10 mL of crude enzyme extract buffered with 200 mmol L$^{-1}$ sodium acetate, pH 5.5. Sodium azide (15 mmol L$^{-1}$) was used to avoid bacterial contamination. In each assay, the sum of the substrate's mixture was always one (SUM = 1). The assays were carried out at 50 °C for 48 h, and aliquots were immediately boiled for 5 min to inactivate the enzymes. The measured response of the hydrolysis rate was assumed to be dependent on the relative proportions of the components in the mixture. Suitable models for mixture designs consist of three components, including linear, quadratic, and special cubic models [37–40]. Therefore, the best-fitting mathematical model was selected based on the comparisons of several statistical parameters, including the coefficient of determination $R^2$ and the F value provided by ANOVA. The most significant model for each studied response was used to plot the triangular surfaces.

### 2.8. HPLC Analysis of the Saccharification Products

One milliliter of the saccharification assays was filtered on a cellulose nitrate membrane (0.45 μm). Thirty microliters of each filtered sample were injected into the Waters HPLC system using a Supelcogel C611 column (7.8 × 300 mm; 60 °C) and eluted with 0.4 mmol L$^{-1}$ NaOH as the mobile phase at a flow rate of 0.5 mL min$^{-1}$ (controlled by a peristaltic pump). D-glucose and D-xylose were monitored with a differential refractometer (model R401, Waters). They were identified by comparing the retention times of the peaks in the samples against the commercial standards glucose (>95%) and xylose (>95%) (Sigma-Aldrich, St. Louis, MO, USA)

### 2.9. Statistical Analysis and Reproducibility of Experiments

Experimental design and statistical analyses were performed using the Statistica software (Version 13.0, StatSoft, Inc., Tulsa, OK, USA). The experiments performed using the OFAT method were conducted in triplicate. The pure error of the CCD experiments and the MD were calculated based on the values of the center point response variables (minimum of three replicates).

## 3. Results and Discussion

### 3.1. Optimization of Xylanase Production and Response Surface Analysis

The linear, quadratic, and interaction effects of three independent variables ($X_1$, $X_2$, and $X_3$) on xylanase production from *A. tamarii* Kita were investigated through a $2^3$ full-factorial CCD. The chemical compositions of these residues from literature reports are summarized in Supplementary Table S1. The experimental and predicted values of xylanolytic activity and respective residues are reported in Table 1.

ANOVA was performed on the regression model selected, as shown in Table 2. This result demonstrates that the quadratic regression model is highly significant since its F value was 4.94 times higher than the F critical value at a level of significance of 5%. The coefficient of determination ($R^2$), which measures the goodness of fit of a model, indicates that the model can explain 90% of the sample variability. The proposed model can still be considered predictive since its F value for the lack of fit (9.22) was lower than the F critical value (19.37), i.e., it is not significant [38,41,42]. Regarding the effects, only the linear and quadratic terms were considered statistically significant through a hypothesis test that follows a Student's t-distribution.

**Table 2.** Analysis of variance of the model regression for xylanase production.

| Source | SS | DF | MS | F Value | F Critical Value |
|---|---|---|---|---|---|
| Model | 98.716 | 6 | 16.453 | 15.19 | 3.22 |
| Residual | 10.834 | 10 | 1.083 | | |
| Lack of fit | 10.548 | 8 | 1.319 | 9.22 | 19.37 |
| Pure error | 0.286 | 2 | 0.143 | | |
| Total | 109.550 | 16 | | | |

$R^2$ = 0.90; SS, sum of squares; DF, degrees of freedom; MS, mean square; significance level = 95%.

Given these considerations, the mathematical model representing xylanase production was simplified by the elimination of statistically insignificant terms:

$$\text{Xylanolytic activity (U mL}^{-1}) = 23.01 + 1.81X_1 - 1.10X_1{}^2 + 0.52X_2 - 0.49X_2{}^2 + 1.01X_3 - 1.68X_3{}^2 \tag{1}$$

The response surface plots described by second-order polynomial equations were generated to determine each variable's optimal level for xylanase production (Figure 1b,d,f). These representations were formulated for different combinations of two factors at the time, while the third factor was kept constant. The spectral graphs (Figure 1a,c,e) facilitate understanding each response surface generated, showing the enumerated assays of the CCD matrix (Table 1). According to the desirability profile, which detailed the precise proportions of the independent variables [41] (Figure 2), the highest predicted value of

xylanase for the model (23.512 U mL$^{-1}$) would be obtained with 0.814 g (+0.84) of BB (X$_1$), 20 mL (+0.84) of Adam's' medium (X$_2$), and 144 h (+0.84) of culture time (X$_3$). The highest value obtained with CCD (23.302 U mL$^{-1}$ of xylanase) was observed in assay no. 10 (central point replicate) with 0.625 g of BB, 15 mL of Adams' medium, and 120 h of culture time). Therefore, validation of the experimental model was carried out by conducting additional experiments under the conditions of the central point (assays 18, 19, and 20; Table 1). This combination led to xylanolytic activity of 23.122 ± 0.306 U mL$^{-1}$ (mean), comparable to the model-predicted value (23.011 U mL$^{-1}$). Thus, it can be concluded that this regression model has excellent predictive capacity for xylanase production.

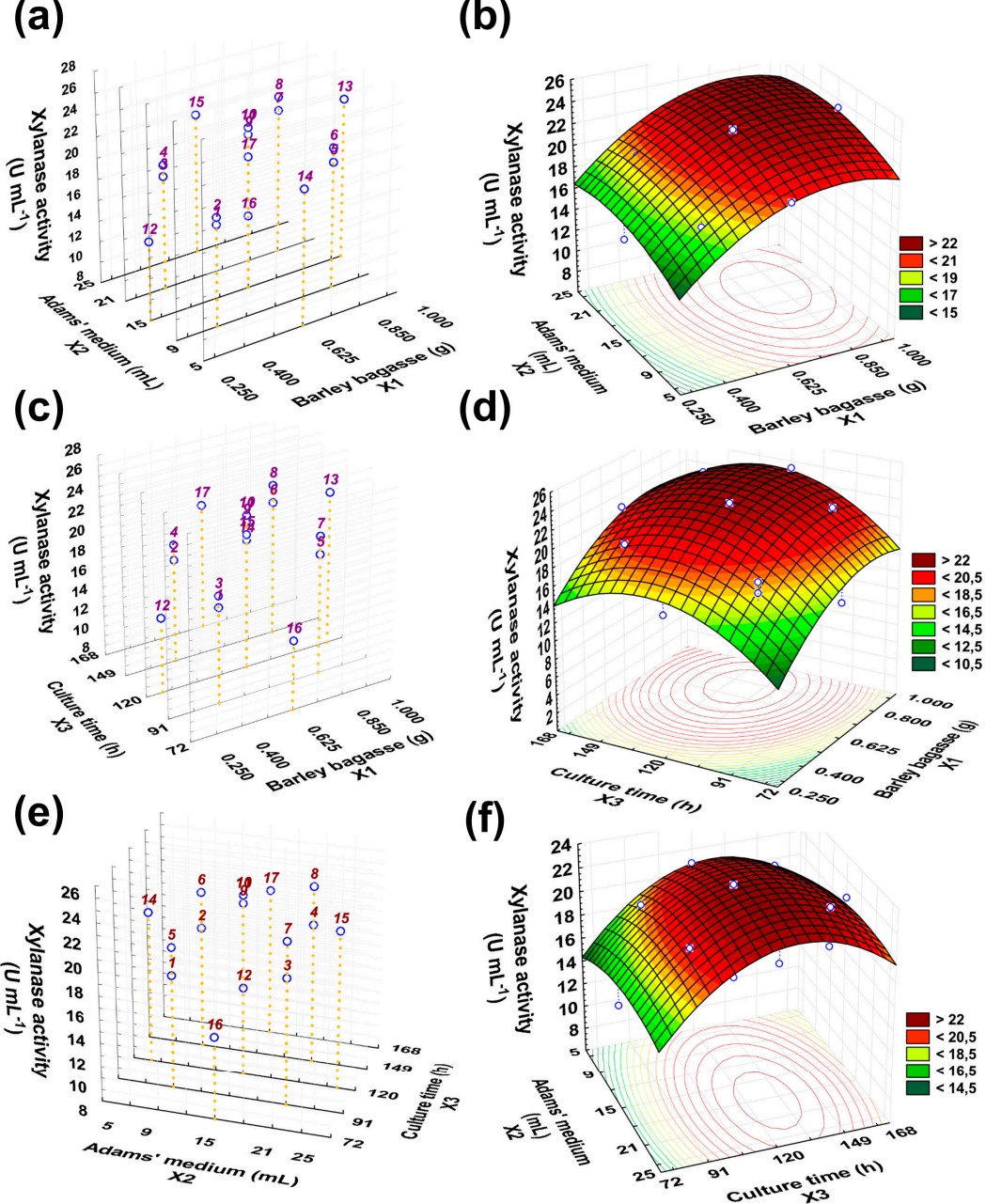

**Figure 1.** Spectral graphs of the CCD matrix (**a**,**c**,**e**) and response surface plots of xylanase production from *A. tamarii* Kita (**b**,**d**,**f**) show the interaction between barley bagasse and volume of Adams' medium (**a**,**b**), barley bagasse and culture time (**c**,**d**) and culture time and volume of Adams' medium (**e**,**f**). The numbers in the spectral graphs (**a**,**c**,**e**) refer to the assay's numbers in Table 1.

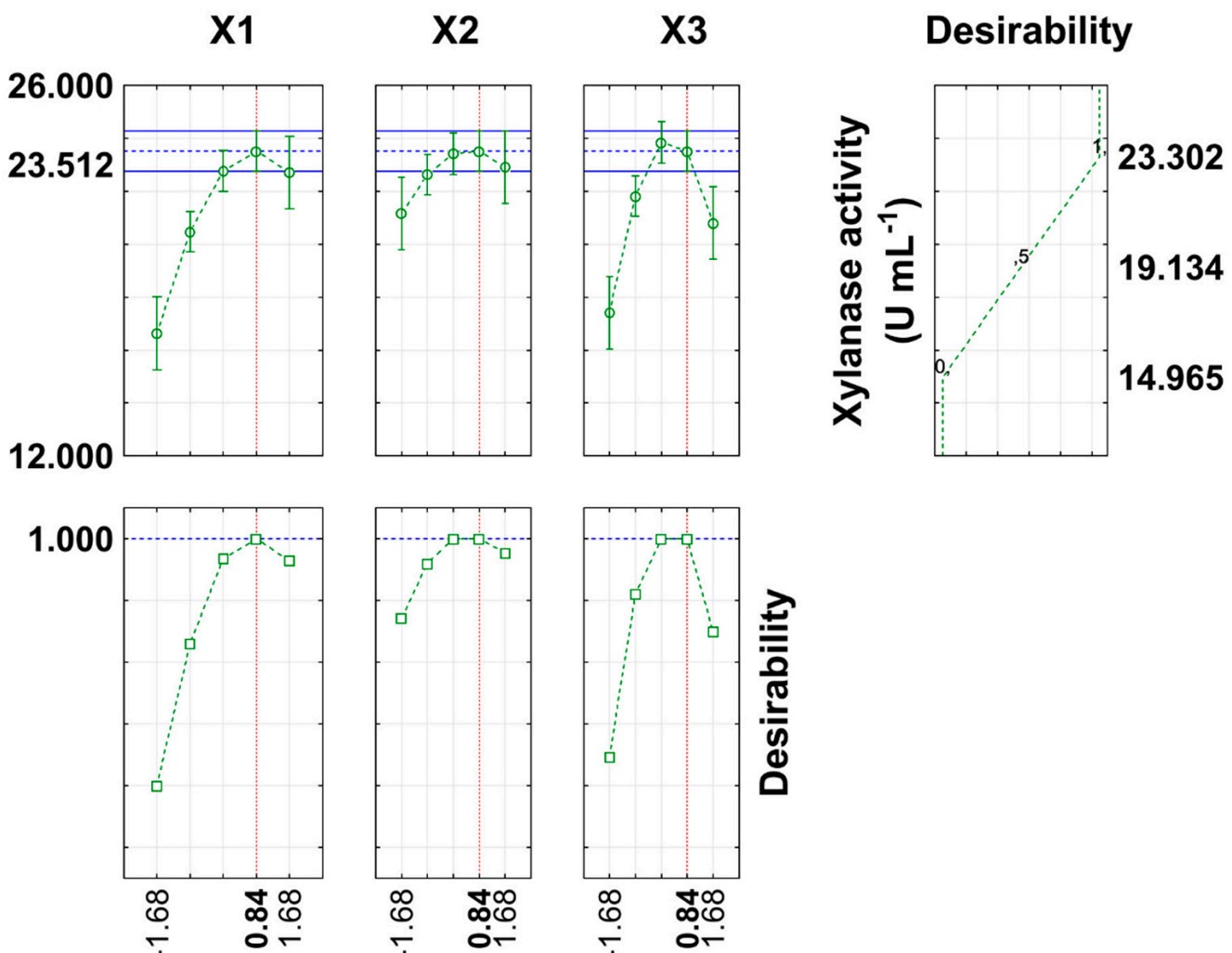

**Figure 2.** Desirability profile for xylanase activity (U mL$^{-1}$). $X_1$ (Barley bagasse (BB)), $X_2$ (Adams' medium), and $X_3$ (Culture time). Red lines indicate the optimization values. Green dashed lines indicate the experimental values. Blue dashed lines indicate the highest predicted value of xylanase activity. Blue horizontal lines show the ±95% confidence interval of the predicted value.

### 3.2. Zymogram Analysis

In order to promote the classification of these xylanases, the equivalent protein bands to xylanase activity positions were excised from non-denaturing PAGE gel and analyzed by MALDI-TOF/TOF. The engine Mascot (Version 2.3, Matrix Science Ltd., London, UK) was used to compare the MS/MS profiles against predicted protein sequences in the NCBI non-redundant protein sequences database (RefSeq). Using the peptide fragments obtained from xylanases 1 and 2 (xylanase 1: GQVTSDGGTYNIYTSVR; xylanase 2: DSVFSQVLGEDFVR) suggests that xylanase 1 is a member of the glycoside hydrolase (GH) 11 family, with xylanase 2 being a member of the GH 10 family. Because of their physicochemical characteristics, GH 10 xylanases typically present a molecular weight greater than 30 kDa and acidic pI values. In contrast, GH 11 xylanases have molecular weights of less than 20 kDa and alkaline pI values [8,43]. Under these particular experimental conditions, the limited electrophoretic mobility of xylanase 1 in non-denaturing PAGE for acid proteins (Figure 3) could be attributed to the pKa of the ionizable side-chains. Similarly, Xiao et al. [44] also reported the simultaneous production of GH 10 and GH 11 endo-xylanases in *Rhizopus oryzae*.

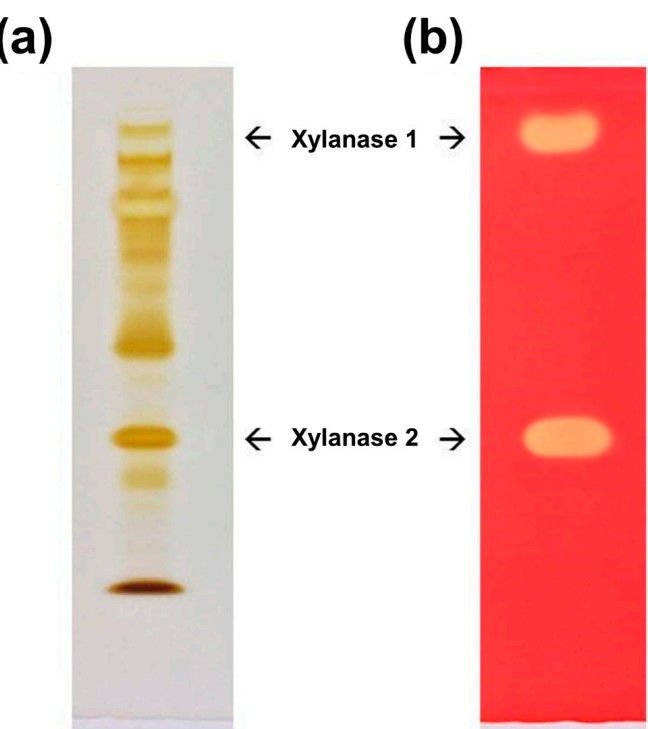

**Figure 3.** (**a**) Non-denaturing PAGE and (**b**) zymogram of the xylanolytic activities from *A. tamarii* Kita crude extract.

### 3.3. Quantification of Enzyme Activities in the Crude Extract

The endo-β-1,4-xylanase, β-xylosidase, and α-L-arabinofuranosidase activities in the optimized crude extract were 23.122 U mL$^{-1}$ ± 0.313, 0.025 U mL$^{-1}$ ± 0.002 and 0.243 U mL$^{-1}$ ± 0.012, respectively (Supplementary Table S2). These enzymes are part of the xylanolytic system, which mainly releases xylose and xylooligosaccharides from substrates, as well as other chemical compounds (glucuronic acid, acetyl, and methyl groups) which can be applied in various chemical industries, including the production of second-generation bioethanol [45].

Although this study focused on xylanase production, Supplementary Table S2 shows that other enzymes were produced during the submerged fermentation. Amylase activity (6.5 U mL$^{-1}$ ± 0.158) was the second-highest enzymatic activity found. Amylases are hydrolytic enzymes that catalyze the hydrolysis of starch in lower molecular weight carbohydrates and have applications in many industrial processes [33,46]. On the other hand, very low levels of endo-1,4-β-glucanase (0.150 U mL$^{-1}$ ± 0.014), cellobiohydrolase (0.255 U mL$^{-1}$ ± 0.015), and 1,4-β-glucosidase (0.040 U mL$^{-1}$ ± 0.003) activities were detected. These results are exciting, since cellulase-free xylanases represent an important enzymatic strategy to access the cellulosic material in the pulp and paper industries, maintaining the integrity of the cellulose microfibrils [47,48].

### 3.4. Mixture Design

For fermentable sugar production, such as glucose and xylose, combinations of three agro-industrial residues were studied in nine assays using an MD strategy. Based on the results presented in Table 3, it can be concluded that ternary blending allowed a more pronounced release of xylose and glucose as compared to the isolated use of the SCBn and SCBp residues, although the higher levels of total reducing sugars (2.802 mg mL$^{-1}$), glucose (0.466 mg mL$^{-1}$), and xylose (0.368 mg mL$^{-1}$) were detected with the use of BB alone.

**Table 3.** Matrix of the simplex centroid mixture design for simultaneous production of different sugars from enzymatic saccharification.

| Run | Independent Variables [a] | | | Responses | | |
| | BB (A) | SCBn (B) | SCBp (C) | Total Reducing Sugars (mg mL$^{-1}$) [b] | Glucose (mg mL$^{-1}$) [c] | Xylose (mg mL$^{-1}$) [c] |
|---|---|---|---|---|---|---|
| 1 | 1 | 0 | 0 | 2.802 | 0.466 | 0.368 |
| 2 | 0 | 1 | 0 | 0.565 | 0.055 | 0.129 |
| 3 | 0 | 0 | 1 | 0.901 | 0.072 | 0.357 |
| 4 | 0.5 | 0.5 | 0 | 1.611 | 0.242 | 0.232 |
| 5 | 0.5 | 0 | 0.5 | 2.061 | 0.134 | 0.235 |
| 6 | 0 | 0.5 | 0.5 | 1.198 | 0.097 | 0.314 |
| 7 | 0.333 | 0.333 | 0.333 | 1.817 | 0.287 | 0.361 |
| 8 | 0.333 | 0.333 | 0.333 | 1.802 | 0.294 | 0.369 |
| 9 | 0.333 | 0.333 | 0.333 | 1.885 | 0.288 | 0.369 |

[a] BB (in natura barley bagasse), SCBn (in natura sugarcane bagasse), and SCBp (steam-exploded sugarcane bagasse). [b] Analysis performed by theMiller method. [c] Analysis performed in HPLC.

According to the statistical parameters of the fitted models (see Table 4), the quadratic model best described the release of total reducing sugars. In contrast, the special cubic model best represents glucose and xylose release. All models were significant at a confidence level of 95% and presented $R^2$ values close to 1, having a good quality fit of the regressions and the experimental data. Moreover, as there is no evidence of lack of fit ($F_{calculated} < F_{tabulated}$), the models could be satisfactorily employed to predict the response variable yield, find the optimum formulations, and improve the performance of the mixtures [49]. Among the independent variables, (A) BB was the substrate with the highest significant effect on the three variable responses studied, followed in order of importance by (C) SCBp and (B) SCBn. The binary interactions of BB with SCBp and SCBn with SCBp presented a positive effect on the production of total reducing sugars, and the ternary interaction presented the highest positive effect on glucose and xylose production. The BB interaction with SCBn was insignificant for any of the responses studied, and the coefficient for this interaction did not contribute to the regression equations (Table 4).

**Table 4.** F-values of lack of fit and regression models, $R^2$ values, and equations of the final reduced models for total reducing sugars, glucose, and xylose productions.

| Response | Lack of Fit | | Model | $R^2$ | Regression | | Equations |
| | $F_{calc}$ | $F_{tab}$ | | | $F_{calc}$ | $F_{tab}$ | |
|---|---|---|---|---|---|---|---|
| Total reducing sugars (mg mL$^{-1}$) | 5.56 | 19.00 | Quadratic | 0.993 [a] | 136.91 | 6.39 | Y = 2.789A + 0.552B + 0.884C + 1.174AC + 2.196BC |
| Glucose (%) | 15.92 | 18.51 | Special Cubic | 0.998 | 342.86 | 9.01 | Y = 0.460A + 0.049B + 0.072C − 0.528AC + 0.146BC + 3.739ABC |
| Xylose (%) | 8.51 | 18.51 | Special Cubic | 0.996 | 158.05 | 9.01 | Y = 0.363A + 0.124B + 0.357C − 0.499AC + 0.295BC + 2.916ABC |

(A) (BB, in natura barley bagasse), (B) (SCBn, in natura sugarcane bagasse), and (C) (SCBp, steam-exploded sugarcane bagasse). [a] Significant at a confidence level of 95%.

Regarding the desirability profile [41,50] for the yield of reducing sugars (mg mL$^{-1}$), the best condition (2.802 mg mL$^{-1}$) was obtained with 100% BB (desirability of 99.43%). This was also observed for the glucose yield (%), with the maximum predicted by the model equal to 0.46% (98.50% desirability). For xylose, the best-predicted yield (0.37%; the desirability of 99.99%) was predicted for a ternary mixture containing the following proportion: 25% BB, 32% in natura sugarcane bagasse (SCBn), and 43% steam-exploded sugarcane bagasse (SCBp).

The goal in MD is to determine if there is a blend that produces a more optimal response (whether this is to maximize or minimize some property) [51]. In our case, the common agro-industrial residues were used as a source of glucose and xylose (SCBn and

SCBp), and the BB was used as a carbon source to grow the *A. tamarii* Kita. In these conditions, the enzymes from fungal crude extract released more reducing sugars, glucose, and xylose from BB, which are good sources of these sugars (without blending with other agro-industrial residues). The BB mainly consists of fiber (cellulose and hemicellulose), protein, and lignin, with the content varying depending on the harvesting time, cereal variety, type of hops added, and the malting and mashing regime, including whether adjuncts were employed during brewing [52]. The efficiency of mashing could increase the content of starchy endosperm and aleurone cells in the BB reaching 12% of dry weight of starch. Non-starch polysaccharides comprise 30–50% of the dry weight, with 20–25% hemicellulose (primarily arabinoxylan) and 12–25% cellulose. Additionally, proteins constitute 19–30%, lignin accounts for 12–28%, lipids contribute 10%, and ashes represent 2–5% [52]. Moreira et al. [53] demonstrated that *A. tamarii* produces constitutive glucoamylase and $\alpha$-amylase under static conditions using Vogel media, even without an additional carbon source. This constitutive amylase production could also account for the enzyme activity observed in the crude extract. (6.5 U mg$^{-1}$). Previously, the composition of BB was determined by our group, showing 33.7% cellulose, 39.7% hemicellulose, 14.3% lignin, 11.7% pectin, and 0.6% starch [41]. In this scenario, even if the starch content in BB was negligible, the *A. tamarii* could produce amylases. Mixtures of glycosyl hydrolases from *Aspergillus japonicus*, *Aspergillus versicolor,* and *Trichoderma reesei* released more than 42% of the total polysaccharide sugars from BB in one day. It was shown that non-cellulosic glucose, part of the feruloylated arabinoxylan, and over 50% of the protein content could be released from BB when treated with microbial commercial proteases in 24 h [54–56]. This non-cellulosic glucose could also account for the higher glucose content in Run 1 compared to SCBn and SCBp since *A. tamarii* has been reported as an efficient producer of proteases [23,24].

Contour Plots of Sugar Production

Mixture contour plots (Figure 4) show the variation in the amount of sugars produced by different proportions of agro-industrial residues. Each residue is represented in the corner of an equilateral triangle. Each point within this triangle refers to a different proportion of components in the mixture, where the sum of the components is unitary. Furthermore, the triangular surface displays a two-dimensional view where all points with the same response are connected to produce contour lines of constant responses (Figure 4a,d,g).

The highest responses for total reducing sugars (Figure 4a,b) and glucose (Figure 4d,e) were located toward the lower left vertex of their triangular surfaces, corresponding to the composition of BB residue only. Therefore, the release of total reducing sugars and glucose was better when higher proportions of BB were present in the mixture. This higher glucose yield with BB could be related to the action of amylases on residual starch fragments in this biomass [57]. In the case of sugarcane residues (Runs 2–3 in Table 3), the lower glucose yield could be related to the reduced abundance of starch polysaccharides and/or the low cellulase activity of *A. tamarii* Kita enzymatic extract as observed in Supplementary Table S2. Run 3, with only SCBp, released more glucose, xylose, and total reducing sugars compared to Run 2 (SCBn). This could be related to the greater accessibility to PCWDEs, since the steam explosion treatment disrupts the lignin structure of sugarcane bagasse [2]. In the crude extract, we observed the presence of 1,4-β-xylosidase (0.281 U mg$^{-1}$ ± 0.022) and 1,4-β-glucosidase (0.449 U mg$^{-1}$ ± 0.034). These enzymes are responsible for releasing xylose and glucose, respectively (Supplementary Table S2).

Figure 4g,h indicate high production of xylose over a broad range of lignocellulose mixtures, including the ternary combination at the center of the triangle. The xylose released from the assay with SCBp only (Run 3 in Table 3) was higher when compared to SCBn only (Run 2 in Table 3). This could be related to the lower structural recalcitrance of sugarcane bagasse after steam-explosion pretreatment. According to Gao et al. [58], a large portion of lignin and xylan is removed from sugarcane bagasse after this pretreatment process (Supplementary Table S1) [58,59]. The variation in the quality adjustment is shown in the graphs of predicted versus observed values: the model for xylose (%) had a reasonable fit

(Figure 4i). However, the models for reducing sugars (mg mL$^{-1}$) (Figure 4c) and glucose (%) (Figure 4f) appear to have a good predictive performance. The combination of the three agro-industrial residues seems to be a viable strategy when aiming to increase the yield of glucose and xylose using SCBn and SCBp (Figure 4d,g, respectively), and where adequate modeling (especially in cubic models) highlighted the synergistic effect between the mixture components.

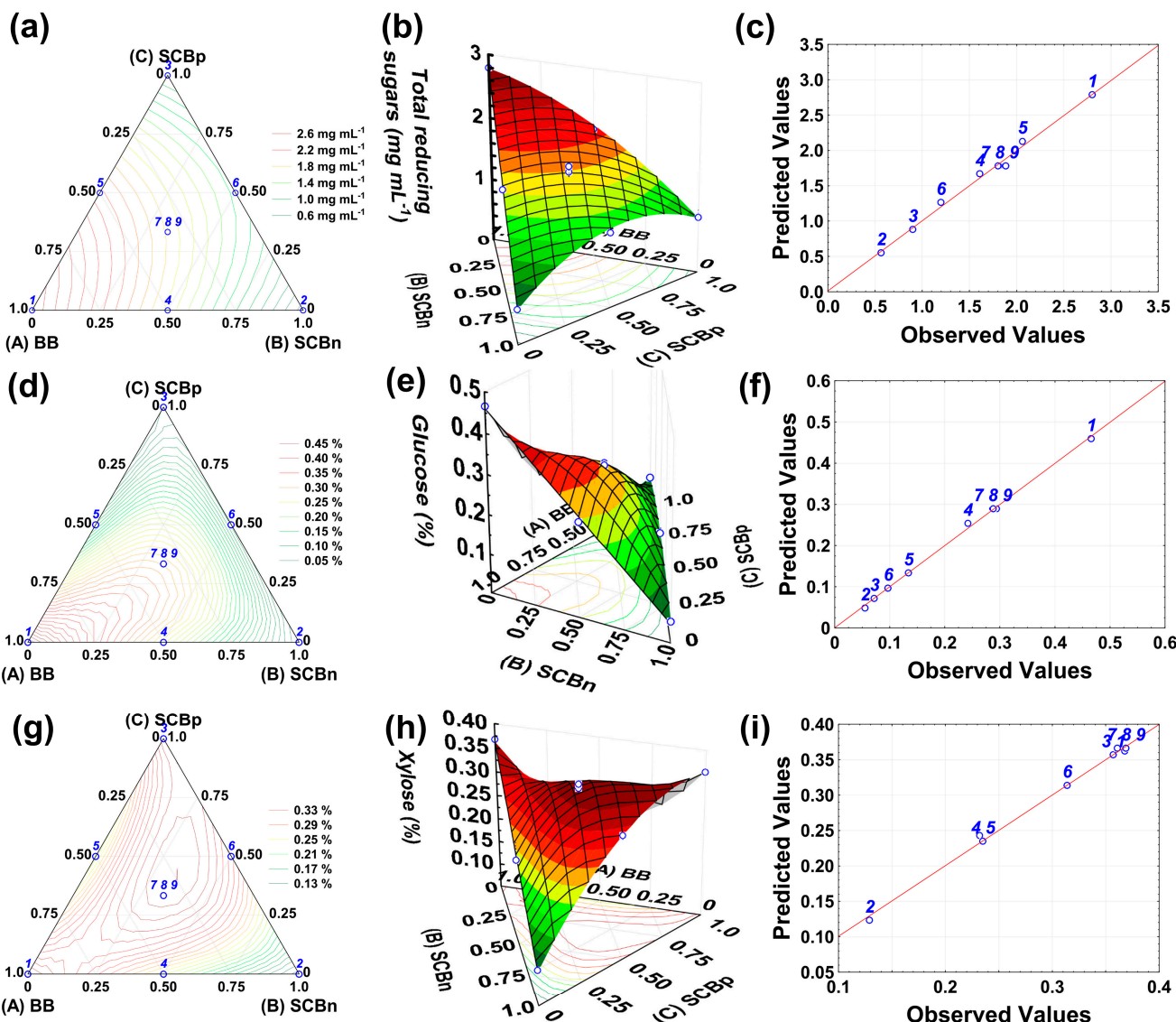

**Figure 4.** Contour plots and surface response plots from mixture design experiments showing the interaction between the three agro-industrial residues for the production of fermentable sugars after 48 h of hydrolysis: reducing-sugar yield (mg mL$^{-1}$) (**a,b**); glucose yield (%) (**d,e**); xylose yield (%) (**g,h**). The variation in the quality of the fit is revealed by the plot of the predicted responses against the observed responses: reducing sugar yield (mg mL$^{-1}$) (**c**); glucose yield (%) (**f**); xylose yield (%) (**i**). The numbers in the graphs (**a,c,d,f,g,i**) refer to the assay's numbers in Table 3.

## 4. Conclusions

*Aspergillus tamarii* Kita produces high xylanase levels in Adams' culture medium supplemented with BB, a by-product of the brewing industry. Its enzyme extract exhibited two xylanase activities in the zymogram identified by mass spectrometry as glycosyl hydrolases belonging to families 10 and 11 (GH10 and GH11). The CCD showed an excellent predictive capacity for xylanase production. In addition, MD results indicated that

enzymatic saccharification of three different lignocellulosic residues could be an exciting bioremediation technique for rejected agro-industrial lignocellulosic materials, as well as being useful for the simultaneous release of glucose and xylose by *A. tamarii* Kita crude enzyme extract.

**Supplementary Materials:** The following are available online at https://www.mdpi.com/article/10.3390/fermentation10050241/s1, Supplementary Table S1—Glucan, xylan, and Klason lignin relative composition; and Supplementary Table S2—Enzyme activities present in the crude extract of *A. tamarii* Kita.

**Author Contributions:** Conceptualization, M.d.L.T.d.M.P. and R.J.W.; Methodology, J.C.S.S., P.R.H., J.M.M. and L.M.O.-M.; Validation, J.C.S.S., P.R.H., J.M.M. and L.M.O.-M.; Investigation, J.C.S.S., P.R.H., J.M.M. and C.G.V.R.; Data Curation, J.M.M., C.G.V.R., A.M., M.K.K. and M.d.L.T.d.M.P.; Writing—Original Draft Preparation, J.C.S.S., P.R.H. and J.M.M.; Writing—Review and Editing, J.C.S.S., P.R.H., M.C. and M.d.L.T.d.M.P.; Visualization, M.d.L.T.d.M.P. and R.J.W.; Supervision, M.d.L.T.d.M.P. and R.J.W. Project Administration, M.d.L.T.d.M.P. All authors have read and agreed to the published version of the manuscript.

**Funding:** This work was supported by grants from the Fundacão de Amparo à Pesquisa do Estado de São Paulo (FAPESP) nos. 2019/21989-7, 2018/07522-6, 2023/01547-5; Conselho de Desenvolvimento Científico e Tecnológico (CNPq) nos. 310340/2021-7, 310840/2021-0, 384465/2023-4, and Coordenação de Aperfeiçoamento de Pessoal de Nível Superior (CAPES) Financial Code 001. This project is also part of the National Institute of Science and Technology of Bioethanol (INCT-Bioetanol) (FAPESP 2014/50884-5/CNPq 465319/2014-9).

**Institutional Review Board Statement:** Not applicable.

**Informed Consent Statement:** Not applicable.

**Data Availability Statement:** Data are contained within the article and Supplementary Materials.

**Acknowledgments:** We thank Mauricio de Oliveira for the technical assistance.

**Conflicts of Interest:** The authors declare no conflict of interest.

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
