# Peer review of "Enhancing Xylanase Production from Aspergillus tamarii Kita and Its Application in the Bioconversion of Agro-Industrial Residues into Fermentable Sugars Using Factorial Design"

_fermentation, doi:10.3390/fermentation10050241_

Round 1
Reviewer 1 Report
Comments and Suggestions for Authors
This manuscript is not at all in a good position to be accepted in any journal. There are many fundamental problems in this manuscript and some of them are enlisted here:
1. Introduction is not appropriate and needs improvement. For instance, the paragraph "The increasing interest in xylanases..." should have been placed at the beginning of the manuscript to emphasize the significance of xylanases. The manuscript predominantly focuses on microbial fermentation for hemicellulase production, yet this is not evident in the Introduction section. There is a lack of clarity regarding which microorganisms are capable of producing hemicellulase and which compounds induce hemicellulase production. Additionally, the rationale for using barley bagasse as an inducer is not explained.
2. Why was the strain Aspergillus tamarii chosen for this study? Has this particular strain been reported in previous research?
3. In the CCD experiment, why was Barley bagasse chosen, along with Adam's medium and the specific culture time? Shouldn't pH and temperature be considered as well? Furthermore, large-scale hemicellulase fermentation typically involves fed-batch fermentation or solid-state fermentation. Is this CCD model appropriate for large-scale production?
4. The manuscript mentions, "These results are very interesting since cellulase-free xylanases represent an important enzymatic strategy to access the cellulosic material in the pulp and paper industries, maintaining the integrity of the cellulose microfibrils." However, glucose was detected in the hydrolysis experiments.
5. Results have not been compared with other reported hemicellulase production data.
6. Did the Zymogram Analysis show only two hemicellulases and no β-xylosidase? If not, why did the hydrolysis experiment result in xylose production?
Author Response
Ribeirão Preto, April 10th, 2024
Ms. Jelena Licinar
Assistant Editor, MDPI Belgrade
e-mail: licinar@mdpi.com
MDPI, Fermentation Editorial Office
St. Alban-Anlage 66, 4052 Basel, Switzerland
E-Mail: fermentation@mdpi.com
Reference: Manuscript ID: fermentation-2942792
I have attached the revised manuscript entitled “Enhancing xylanase production from Aspergillus tamarii Kita and its application in the bioconversion of agro-industrial residues into fermentable sugars using factorial design.” We appreciate you and the reviewers' time spent revising this manuscript. The modifications are in Red in the text. We also would like to express our gratitude for the pertinent comments you have made on our work.
We look forward to hearing from you.
Yours sincerely,
Maria de Lourdes T. M. Polizeli, PhD
Corresponding author
Reviewer 1
- Introduction is not appropriate and needs improvement. For instance, the paragraph "The increasing interest in xylanases..." should have been placed at the beginning of the manuscript to emphasize the significance of xylanases. The manuscript predominantly focuses on microbial fermentation for hemicellulase production, yet this is not evident in the Introduction section. There is a lack of clarity regarding which microorganisms are capable of producing hemicellulase and which compounds induce hemicellulase production. Additionally, the rationale for using barley bagasse as an inducer is not explained.
A: Thanks for the suggestion. The introduction was modified as suggested.
- Why was the strain Aspergillus tamarii chosen for this study? Has this particular strain been reported in previous research?
A: The saprophytic fungus Aspergillus tamarii has been recognized as an efficient producer of proteases (Boer et al., 2000; Anandan et al., 2007), xylanases (El-Gindy et al, 2015, Ferreira et al, 1999, Monclaro et al 2016, Gouda et al, 2002) and various enzymes from our scientific group (Heinen et al 2017; Heinen et al 2018). This information was added to the introduction in the revised manuscript.
*The references can be found in a separate section at the end of the answers
- In the CCD experiment, why was Barley bagasse chosen, along with Adam's medium and the specific culture time? Shouldn't pH and temperature be considered as well? Furthermore, large-scale hemicellulase fermentation typically involves fed-batch fermentation or solid-state fermentation. Is this CCD model appropriate for large-scale production?
A: The barley bagasse, Adam’s medium, and culture times were chosen because this fungus had previously been studied by our research group using the One-Factor-At-a-Time (OFAT) approach, with the goal of purifying xylanases (Heinen et al 2017; Heinen et al 2018). The focus of the CCD in this manuscript was to evaluate the optimal conditions for barley bagasse and Adam’s medium in order to optimize the production of plant cell wall-degrading enzymes (PCWDEs) and utilize them for biomass enzymatic hydrolysis. Further studies should be conducted to assess the viability of large-scale fermentation.
*The references can be found in a separate section at the end of the answers
- The manuscript mentions, "These results are very interesting since cellulase-free xylanases represent an important enzymatic strategy to access the cellulosic material in the pulp and paper industries, maintaining the integrity of the cellulose microfibrils." However, glucose was detected in the hydrolysis experiments.
A: The BB is mainly constituted of fiber (cellulose and hemicellulose), protein, and lignin, with the content varying depending on the time of harvesting, cereal variety, type of hops added, the malting and mashing regime, and whether adjuncts were employed during brewing (Lynch et al, 2016) The efficiency of mashing could increase the content of starchy endosperm and aleurone cells on the BB reaching 12 % of dry weight of starch. The non-starch polysaccharides represent 30-50% of dry weight (20-25% hemicellulose – mainly arabinoxylan and 12-25% of cellulose), proteins constitute 19-30%, lignin 12-28%, lipids 10%, and ashes 2-5% (Lynch et al, 2016). Moreira et al. (1999) showed that A. tamarii produces constitutive glucoamylase and α-amylase in static conditions on Vogel media without additional carbon source (15 U mg-1 ±2 of glucoamylase plus α-amylase). This also could explain the amylase activity found in the crude extract, Table 2 (6.5 U mg-1). In this scenario, even if the starch content in BB was negligible, the A. tamarii could produce amylases. Mixtures of glycosyl hydrolases from Aspergillus japonicus, Aspergillus versicolor, and Trichoderma reesei released more than 42% of the total polysaccharide sugars from BB in one day. It was shown that non-cellulosic glucose, part of the feruloylated arabinoxylan, and over 50% of the protein content could be released from BB when treated with microbial commercial proteases in 24 hours (Khan et al, 1988, Faulds et al, 2009, Mussatto 2014). This non-cellulosic glucose could also account for the higher glucose content in Run 1 compared to SCBn and SCBp since A. tamarii has been reported as an efficient producer of proteases (Anandan et al. 2007 and Boer et al. 2000).
*The references can be found in a separate section at the end of the answers
- Results have not been compared with other reported hemicellulase production data.
A: This was improved in the revised manuscript in the Results and Discussion section in the revised manuscript (in red)
- Did the Zymogram Analysis show only two hemicellulases and no β-xylosidase? If not, why did the hydrolysis experiment result in xylose production?
A: The zymogram using Congo Red is more specific to xylanases. In this case, the dye interacts with xylan but not with xylooligosaccharides and xylose. In the supplementary Table 2 of the original submission, 1,4-β-xylosidase activity was reported (0.281 U·mg⁻¹ ±0.022), which could potentially explain the xylose production.
References
Anandan, D.; Marmer, W.N.; Dudley, R.L. Isolation, Characterization and Optimization of Culture Parameters for Production of an Alkaline Protease Isolated from Aspergillus tamarii. J Ind Microbiol Biotechnol 2007, 34, 339–347, doi:10.1007/s10295-006-0201-5.
Boer, C.G.; Peralta, R.M. Production of Extracellular Protease by Aspergillus tamarii. Journal of Basic Microbiology 2000, 40, 75–81, doi:10.1002/(SICI)1521-4028(200005)40:2<75::AID-JOBM75>3.0.CO;2-X.
El-Gindy, A.A.; Saad, R.R.; Fawzi, E.M. Purification of β-Xylosidase from Aspergillus tamarii Using Ground Oats and a Possible Application on the Fermented Hydrolysate by Pichia Stipitis. Ann Microbiol 2015, 65, 965–974, doi:10.1007/s13213-014-0940-x.
Faulds, C.B.; Collins, S.; Robertson, J.A.; Treimo, J.; Eijsink, V.G.H.; Hinz, S.W.A.; Schols, H.A.; Buchert, J.; Waldron, K.W. Protease-Induced Solubilisation of Carbohydrates from Brewers’ Spent Grain. Journal of Cereal Science 2009, 50, 332–336, doi:10.1016/j.jcs.2009.01.004.
Ferreira, G.; Boer, C.G.; Peralta, R.M. Production of Xylanolytic Enzymes by Aspergillus tamarii in Solid State Fermentation. FEMS Microbiology Letters 1999, 173, 335–339, doi:10.1111/j.1574-6968.1999.tb13522.x.
Gouda, M.K.; Abdel-Naby, M.A. Catalytic Properties of the Immobilized Aspergillus tamarii Xylanase. Microbiol Res 2002, 157, 275–281, doi:10.1078/0944-5013-00165.
Heinen, P.R.; Bauermeister, A.; Ribeiro, L.F.; Messias, J.M.; Almeida, P.Z.; Moraes, L.A.B.; Vargas-Rechia, C.G.; de Oliveira, A.H.C.; Ward, R.J.; Filho, E.X.F.; et al. GH11 Xylanase from Aspergillus tamarii Kita: Purification by One-Step Chromatography and Xylooligosaccharides Hydrolysis Monitored in Real-Time by Mass Spectrometry. International Journal of Biological Macromolecules 2018, 108, 291–299, doi:10.1016/j.ijbiomac.2017.11.150.
Heinen, P.R.; Pereira, M.G.; Rechia, C.G.V.; Almeida, P.Z.; Monteiro, L.M.O.; Pasin, T.M.; Messias, J.M.; Cereia, M.; Kadowaki, M.K.; Jorge, J.A.; et al. Immobilized Endo-Xylanase of Aspergillus tamarii Kita: An Interesting Biological Tool for Production of Xylooligosaccharides at High Temperatures. Process Biochemistry 2017, 53, 145–152, doi:10.1016/j.procbio.2016.11.021.
Khan, A.W.; Lamb, K.A.; Schneider, H. Recovery of Fermentable Sugars from the Brewers’ Spent Grains by the Use of Fungal Enzymes. Process Biochem 1988, 23, 172–175.
Lynch, K.M.; Steffen, E.J.; Arendt, E.K. Brewers’ Spent Grain: A Review with an Emphasis on Food and Health. Jour-nal of the Institute of Brewing 2016, 122, 553–568, doi:10.1002/jib.363.
Monclaro, A.V.; Aquino, E.N.; Faria, R.F.; Ricart, C. a. O.; Freitas, S.M.; Midorikawa, G.E.O.; Miller, R.N.G.; Michelin, M.; Polizeli, M. de L.T. de M.; Filho, E.X.F. Characterization of Multiple Xylanase Forms from Aspergillus tamarii Re-sistant to Phenolic Compounds. Mycosphere 2016, 7, 1554–1567, doi:10.5943/mycosphere/si/3b/7.
Moreira, F.G.; Lima, F.A. de; Pedrinho, S.R.F.; Lenartovicz, V.; Souza, C.G.M. de; Peralta, R.M. Production of Amyla-ses by Aspergillus tamarii. Rev. Microbiol. 1999, 30, 157–162, doi:10.1590/S0001-37141999000200014
Mussatto, S.I. Brewer’s Spent Grain: A Valuable Feedstock for Industrial Applications. J Sci Food Agric 2014, 94, 1264–1275, doi:10.1002/jsfa.6486.
_____________________________________________________
Prof. Dr. Maria de Lourdes T. M. Polizeli
Full Professor
University of São Paulo
Faculty of Philosophy, Sciences and Letters of Ribeirão Preto
Department of Biology
Phone.: +55 16 3315-4680
www.ffclrp.usp.br

Reviewer 2 Report
Comments and Suggestions for Authors
The analyzed work studies the possibilities of increasing the production of xylanases that can be obtained by fermentation in the solid state from the main residue of beer production. Subsequently, the conditions determined in the production of sugars in three agro-industrial substrates are tested.
The work is well designed and falls within the scope of Fermentation.
A few comments that may improve the manuscript are shown below.
- The substrate used is barley bagasse, is it the same as brewer spent grains?
- Authors have selected three factors to assess the best experiemntal conditions to produce enzymes, barley bagasse substrate (X1), the volume of Adams' medium (X2), 106 and culture time (X3). While from the point of view of methodology, it is a right approach, using an experimental design, please justify why these factors were chosen.
A similar justification could be offered for the selection of the three feedstocks, barley bagasse, sugarcane bagasse and steam-exploded sugarcane bagasse used in the mixture design. What is the main objective of this part of the study? Perhaps the Introduction section could include an extra paragraph to address this issue.
Concerning the results obtaoned form the experimental design, please give more information on how you managed the information obtained from the central points of the design, e.g., were they used for calculation of errors?
The concept of desirability could be also explained in more detail.
Conclusions. According to the authors, "Aspergillus tamarii Kita produces high xylanase levels in Adams culture medium supplemented with barley bagasse residues"; Does this mean that this is a better way (than others) to produce xylanases? A comparison table with other results, based on the use of either the same microorganism or the same raw material (or both) would be useful to assess whether or not this could be the method of choice.
Author Response
Ribeirão Preto, April 10th, 2024
Ms. Jelena Licinar
Assistant Editor, MDPI Belgrade
e-mail: licinar@mdpi.com
MDPI, Fermentation Editorial Office
St. Alban-Anlage 66, 4052 Basel, Switzerland
E-Mail: fermentation@mdpi.com
Reference: Manuscript ID: fermentation-2942792
I have attached the revised manuscript entitled “Enhancing xylanase production from Aspergillus tamarii Kita and its application in the bioconversion of agro-industrial residues into fermentable sugars using factorial design.” We appreciate you and the reviewers' time spent revising this manuscript. The modifications are in Red in the text. We also would like to express our gratitude for the pertinent comments you have made on our work.
We look forward to hearing from you.
Yours sincerely,
Maria de Lourdes T. M. Polizeli, PhD
Corresponding author
Reviewer 2
The analyzed work studies the possibilities of increasing the production of xylanases that can be obtained by fermentation in the solid state from the main residue of beer production. Subsequently, the conditions determined in the production of sugars in three agro-industrial substrates are tested. The work is well designed and falls within the scope of Fermentation. A few comments that may improve the manuscript are shown below.
- The substrate used is barley bagasse, is it the same as brewer spent grains?
A: Yes, in some parts of the world, the term “brewer spent grains” is used, while others prefer the terms “beer bagasse” or “barley bagasse” (Heinen et al 2017; Heinen et al 2018; Monteiro et al, 2021) This is now stated in the revised introduction to be more straightforward.
*The references can be found in a separate section at the end of the answers
- Authors have selected three factors to assess the best experiemntal conditions to produce enzymes, barley bagasse substrate (X1), the volume of Adams' medium (X2), and culture time (X3). While from the point of view of methodology, it is a right approach, using an experimental design, please justify why these factors were chosen.
A: The barley bagasse, Adam’s medium, and culture times were chosen because this fungus had previously been studied by our research group using the One-Factor-At-a-Time (OFAT) approach, with the goal of purifying xylanases (Heinen et al. 2017; Heinen et al 2018). The focus of the CCD in this manuscript was to evaluate the optimal conditions for barley bagasse and Adam’s medium to optimize the production of plant cell wall-degrading enzymes (PCWDEs) and utilize them for biomass enzymatic hydrolysis.
*The references can be found in a separate section at the end of the answers
- A similar justification could be offered for the selection of the three feedstocks, barley bagasse, sugarcane bagasse and steam-exploded sugarcane bagasse used in the mixture design. What is the main objective of this part of the study? Perhaps the Introduction section could include an extra paragraph to address this issue.
A: Thank you. This study aimed to evaluate whether a mixture of different biomass residues (including in natura and exploded sugarcane bagasse) could enhance the release of reducing sugars from the biomass. Mixture experiments fall under the category of particular response surface experiments. In such experiments, the objective is to determine whether a combination of ingredients yields an optimal response (whether that response aims to maximize or minimize a specific property). In our case, the mixture does not lead to an increase in the release of reducing sugars. Instead, the barley bagasse appears to be the best substrate for enzyme hydrolysis when aiming to release higher contents of total reducing sugars, xylose, and glucose. An additional paragraph was added in the introduction of the revised version of this manuscript.
- Concerning the results obtaoned form the experimental design, please give more information on how you managed the information obtained from the central points of the design, e.g., were they used for calculation of errors?
A: Thank you for your suggestion. This was modified in the revised manuscript.
- The concept of desirability could be also explained in more detail.
A: We agree with the suggestion. The concept of desirability was added in the revised manuscript in the Results and Discussion, section 3.1. “Optimization of Xylanase Production and Response Surface Analysis”.
- Conclusions. According to the authors, "Aspergillus tamarii Kita produces high xylanase levels in Adams culture medium supplemented with barley bagasse residues"; Does this mean that this is a better way (than others) to produce xylanases? A comparison table with other results, based on either the same microorganism or the same raw material (or both) would be useful to assess whether or not this could be the method of choice.
A: We thank you for the suggestion, but the results presented in the manuscript are based on our experiments and our group's previous works cited in the text (Heinen et al 2017; Heinen et al 2018). It is indicated that the BB and Adam’s culture medium was suitable for xylanase production. Concerning in this fact, we carried out our experiments.
*The references can be found in a separate section at the end of the answers
References
Heinen, P.R.; Bauermeister, A.; Ribeiro, L.F.; Messias, J.M.; Almeida, P.Z.; Moraes, L.A.B.; Vargas-Rechia, C.G.; de Oliveira, A.H.C.; Ward, R.J.; Filho, E.X.F.; et al. GH11 Xylanase from Aspergillus tamarii Kita: Purification by One-Step Chromatography and Xylooligosaccharides Hydrolysis Monitored in Real-Time by Mass Spectrometry. International Journal of Biological Macromolecules 2018, 108, 291–299, doi:10.1016/j.ijbiomac.2017.11.150.
Heinen, P.R.; Pereira, M.G.; Rechia, C.G.V.; Almeida, P.Z.; Monteiro, L.M.O.; Pasin, T.M.; Messias, J.M.; Cereia, M.; Kadowaki, M.K.; Jorge, J.A.; et al. Immobilized Endo-Xylanase of Aspergillus tamarii Kita: An Interesting Biological Tool for Production of Xylooligosaccharides at High Temperatures. Process Biochemistry 2017, 53, 145–152, doi:10.1016/j.procbio.2016.11.021.
Monteiro, L.M.O.; Vici, A.C.; Messias, J.M.; Heinen, P.R.; Pinheiro, V.E.; Rechia, C.G.V.; Buckeridge, M.S.; de Moraes, M. de L.T. Increased Malbranchea pulchella β-Glucosidase Production and Its Application in Agroindustrial Residue Hydrolysis: A Research Based on Experimental Designs. Biotechnology Reports 2021, 30, e00618, doi:https://doi.org/10.1016/j.btre.2021.e00618.
_____________________________________________________
Prof. Dr. Maria de Lourdes T. M. Polizeli
Full Professor
University of São Paulo
Faculty of Philosophy, Sciences and Letters of Ribeirão Preto
Department of Biology
Phone.: +55 16 3315-4680
www.ffclrp.usp.br

Reviewer 3 Report
Comments and Suggestions for Authors
To manuscript “Enhancing Xylanase Production from Aspergillus tamarii Kita and Its Application in the Bioconversion of Agro-Industrial 3 Residues into Fermentable Sugars Using Factorial Design”:
This research studied the xylanase production from Aspergillus tamarii Kita and further explored its application of three residues for fermentable sugar production. The methodology of
central composite design is sound, and the data collection procedures are well-described. The data presented were informative and good for the xylanase production. However, the introduction could be strengthened by providing more background information on Aspergillus tamarii Kita as well as the reach gap in xylanase production. Also, the results and discussion should be presented in more detail rather than a simple description of a model or result.
Overall, I prefer to give a major revision on this manuscript according to the high standard of Fermentation. I would be happy to reconsider it after these revisions have been made. Here are some suggestions on this manuscript.
1 In the introduction section, does Aspergillus tamarii KitaIs the only microorganism that can produce xylanase? If no, why you choose this microorganism.
2 Please state the current state of the research on the production of xylanase by Aspergillus tamarii KitaIs.
3 Please describe the highlights of this study at the end of the introduction.
4 In line of 108, please make sure the description of 23 is correct.
5 May I know how you extract the enzyme from the microorganism? I didn’t find this part at Material And Methods.
6 Figures 1 and 2 are not clear enough, can you try resubmitting them in high resolution?
7 3.1, you describe a lot to validate the model you have, but ignore how these variables affect xylanase production and why?
8 In line263, why was the amylase activity so high? Does it affect xylanase production? How to control it and to produce as much as xylanase?
9 In line of 274, how can you get the conclusion that “ternary blending allowed a more pronounced release of xylose and glucose as compared to the isolated use of the SCBn and SCBp residues” ? According to Table 3, the data of total reducing sugars, glucose and xylose indicate that barley bagasse alone was the optimum substrate.
10 In line of 293, similar, which data showed “the ternary interaction presented the highest positive effect on glucose and xylose production”?
11Why natura barley bagasse produce so much reduced sugars than pretreated sugarcane bagasse? From my knowledge, it shouldn’t be like this? If the data is correct, does it mean that the enzyme produced by A. tamarii Kita is not suitable for application to sugarcane bagasse?
12 How much starch in your barley bagasse? Since A. tamarii Kita can produce amylase, so, how much sugars (shows in Table 3) come from starch?
13 What is the purpose for doing a substate mixture design?
14 In line of 325, since the BB produced the highest reduced sugars as well as glucose and xylose, it is obvious to get a better sugar yield by improving the proportion of BB in the mixture substrate from Table 3. So, you don’t even have the mixture design part.
15 Utilizing the scientific method to design experiments is important, but it is crucial to remember that the experimental design should serve the data. The data analysis in this case appears to be superficial and lacks depth. I strongly suggest more description in the biological xylanase-producing mechanism and related influencing factors.
Comments on the Quality of English LanguageThe language is good.
Author Response
Ribeirão Preto, April 10th, 2024
Ms. Jelena Licinar
Assistant Editor, MDPI Belgrade
e-mail: licinar@mdpi.com
MDPI, Fermentation Editorial Office
St. Alban-Anlage 66, 4052 Basel, Switzerland
E-Mail: fermentation@mdpi.com
Reference: Manuscript ID: fermentation-2942792
I have attached the revised manuscript entitled “Enhancing xylanase production from Aspergillus tamarii Kita and its application in the bioconversion of agro-industrial residues into fermentable sugars using factorial design.” We appreciate you and the reviewers' time spent revising this manuscript. The modifications are in Red in the text. We also would like to express our gratitude for the pertinent comments you have made on our work.
We look forward to hearing from you.
Yours sincerely,
Maria de Lourdes T. M. Polizeli, PhD
Corresponding author
Reviewer 3
To manuscript “Enhancing Xylanase Production from Aspergillus tamarii Kita and Its Application in the Bioconversion of Agro-Industrial 3 Residues into Fermentable Sugars Using Factorial Design”: This research studied the xylanase production from Aspergillus tamarii Kita and further explored its application of three residues for fermentable sugar production. The methodology of central composite design is sound, and the data collection procedures are well-described. The data presented were informative and good for the xylanase production. However, the introduction could be strengthened by providing more background information on Aspergillus tamarii Kita as well as the reach gap in xylanase production. Also, the results and discussion should be presented in more detail rather than a simple description of a model or result. Overall, I prefer to give a major revision on this manuscript according to the high standard of Fermentation. I would be happy to reconsider it after these revisions have been made. Here are some suggestions on this manuscript.
- 1. In the introduction section, does Aspergillus tamarii Kita Is the only microorganism that can produce xylanase? If no, why you choose this microorganism.
A: Thank you for your suggestion. This question was addressed, and the introduction was modified as suggested.
- Please state the current state of the research on the production of xylanase by Aspergillus tamarii Kita Is. Please describe the highlights of this study at the end of the introduction.
- Thank you for your suggestion. This question was addressed in the introduction of the revised manuscript to accommodate the highlights.
- In line of 108, please make sure the description of 23 is correct.
- Thank you. This question was addressed in the revised manuscript.
- May I know how you extract the enzyme from the microorganism? I didn’t find this part at Material And Methods.
A: This information was described in the original manuscript, specifically in Section 2.1: Microorganism and Culture Conditions, in the last sentence: “After fermentation, the cultures were filtered using Whatman nº 1 filter paper, and the extracellular crude enzyme extracts were dialyzed against distilled water for 12 hours before enzymatic assays.” We apologize if the description was unclear. It is the usual form of commenting about this.
- Figures 1 and 2 are not clear enough, can you try resubmitting them in high resolution?
A: Thank you for your observation. We submitted the original figures in .tiff format with a resolution of 300 dpi. However, we believe that the submission process in the Susy system compresses the images, resulting in a lower resolution for the reviewers. Modifications were performed to attend to your suggestion.
- 3.1, you describe a lot to validate the model you have, but ignore how these variables affect xylanase production and why?
A: This section primarily serves for mathematical analysis. The utilization of barley bagasse and Adam’s medium for the growth of A. tamarii was previously standardized in works conducted by our research group (Heinen et al 2017; Heinen et al 2018). Some discussions regarding the factors that affect enzyme production are in Section 3.4 of the revised manuscript.
*The references can be found in a separate section at the end of the answers
- In line263, why was the amylase activity so high? Does it affect xylanase production? How to control it and to produce as much as xylanase?
A: The content of starch varies depending on the harvesting time, cereal variety, type of hops added, the malting and mashing regime, and whether adjuncts were employed during brewing. So, the BB starch could induce amylase production in A. tamarii. Moreira et al. (1999) showed that A. tamarii produces constitutive glucoamylase and α-amylase in static conditions on Vogel media without an additional carbon source (15 U mg-1 ±2 of glucoamylase plus α-amylase) which also could explain the amylase activity found in the crude extract (6.5 U mg-1). Low-level constitutive production of glucoamylase and α-amylase was also detected in sucrose, cellobiose, glucose, and raffinose cultures. Still, active synthesis of both enzymes (glucoamylase and α-amylase) occurred only during growth on maltose, starch, amylose, amylopectin, and glycogen (Moreira et al, 1999). Using BB with more starch content could increase the production of amylases.
*The references can be found in a separate section at the end of the answers
- In line of 274, how can you get the conclusion that “ternary blending allowed a more pronounced release of xylose and glucose as compared to the isolated use of the SCBn and SCBp residues” ? According to Table 3, the data of total reducing sugars, glucose and xylose indicate that barley bagasse alone was the optimum substrate. In line of 293, similar, which data showed “the ternary interaction presented the highest positive effect on glucose and xylose production”?
A: We know this because of the results from the MD experiment. When BB was mixed with SCBn and SCBp in equal proportions (1/3, 1/3, and 1/3), the xylose release amounts were 0.361 mg mL-1, 0.369 mg mL-1, and 0.369 mg mL-1 for Runs 7, 8, and 9, respectively. These values were nearly identical to the xylose release from BB alone (0.368 mg mL-1 in Run 1). The glucose content was 0.287 mg mL-1, 0.294 mg mL-1, and 0.288 mg mL-1. However, in Run 6 with a mixture of SCBn and SCBp (in equal proportions, 0.5 and 0.5), 0.314 mg mL-1 of xylose was released, while only 0.097 mg mL-1 of glucose was observed. So, if the purpose is to use SCBn and SCBp as substrates, adding BB to the mix is a better approach. However, if BB is widely available, it is better to use it alone.
- Why natura barley bagasse produce so much reduced sugars than pretreated sugarcane bagasse? From my knowledge, it shouldn’t be like this? If the data is correct, does it mean that the enzyme produced by A. tamarii Kita is not suitable for application to sugarcane bagasse?
A: The outcome depends on the specific sugar and enzyme we target. The assay using SCBp alone (Table 3 - Run 3) yielded nearly the same amount of xylose release (0.357 mg mL-1) as BB alone (Table 3 - Run 1) (0.368 mg mL-1). Therefore, it would be pretty plausible to utilize the crude extract of A. tamarii grown on BB to improve xylose release from SCBp, if desired. It has been observed that mixtures of glycosyl hydrolases from Aspergillus japonicus, Aspergillus versicolor, and Trichoderma reesei were capable of releasing over 42% of the total polysaccharide sugars from BB within a single day. Non-cellulosic glucose was released from BB when specific microbial commercial proteases were used. A portion of the feruloylated arabinoxylan and over 50% of the protein were also released (Khan et al, 1988; Faulds et al, 2009; Mussatto, 2014).
*The references can be found in a separate section at the end of the answers
- How much starch in your barley bagasse? Since A. tamarii Kita can produce amylase, so, how much sugars (shows in Table 3) come from starch?
A: The content of starch in our BB was determined in a previous work of our group, being 0.6 % (Monteiro et al, 2021). It is not possible to determine how much of the glucose came from the starch. This uncertainty arises because Supplementary Table 2 reveals other enzyme activities in the crude extract.
*The references can be found in a separate section at the end of the answers
- What is the purpose for doing a substate mixture design?
A: This study aimed to evaluate whether a mixture of different biomass residues (including in natura and exploded sugarcane bagasse) could enhance the release of reducing sugars from the biomass. Mixture experiments fall under the category of special response surface experiments. In such experiments, the objective is to determine whether a combination of ingredients yields an optimal response (whether that response aims to maximize or minimize a specific property). In our case, the mixture does not lead to an increase in the release of reducing sugars. Instead, the barley bagasse appears to be the best substrate for enzyme hydrolysis when aiming to release higher contents of total reducing sugars, xylose, and glucose.
*The references can be found in a separate section at the end of the answers
- In line of 325, since the BB produced the highest reduced sugars as well as glucose and xylose, it is obvious to get a better sugar yield by improving the proportion of BB in the mixture substrate from Table 3. So, you don’t even have the mixture design part.
A: It was evident that BB produced the highest levels of reducing sugars, glucose, and xylose after conducting the mixture design experiments. However, we were unaware of this before performing the experiments reported in the manuscript.
- Utilizing the scientific method to design experiments is important, but it is crucial to remember that the experimental design should serve the data. The data analysis in this case appears to be superficial and lacks depth. I strongly suggest more description in the biological xylanase-producing mechanism and related influencing factors.
A: Thank you for the suggestion. A more detailed discussion on this topic can be found in Section 3.4: Mixture Design of the Revised Manuscript (red color)
References
Faulds, C.B.; Collins, S.; Robertson, J.A.; Treimo, J.; Eijsink, V.G.H.; Hinz, S.W.A.; Schols, H.A.; Buchert, J.; Waldron, K.W. Protease-Induced Solubilisation of Carbohydrates from Brewers’ Spent Grain. Journal of Cereal Science 2009, 50, 332–336, doi:10.1016/j.jcs.2009.01.004.
Heinen, P.R.; Bauermeister, A.; Ribeiro, L.F.; Messias, J.M.; Almeida, P.Z.; Moraes, L.A.B.; Vargas-Rechia, C.G.; de Oliveira, A.H.C.; Ward, R.J.; Filho, E.X.F.; et al. GH11 Xylanase from Aspergillus tamarii Kita: Purification by One-Step Chromatography and Xylooligosaccharides Hydrolysis Monitored in Real-Time by Mass Spectrometry. International Journal of Biological Macromolecules 2018, 108, 291–299, doi:10.1016/j.ijbiomac.2017.11.150.
Heinen, P.R.; Pereira, M.G.; Rechia, C.G.V.; Almeida, P.Z.; Monteiro, L.M.O.; Pasin, T.M.; Messias, J.M.; Cereia, M.; Kadowaki, M.K.; Jorge, J.A.; et al. Immobilized Endo-Xylanase of Aspergillus tamarii Kita: An Interesting Biological Tool for Production of Xylooligosaccharides at High Temperatures. Process Biochemistry 2017, 53, 145–152, doi:10.1016/j.procbio.2016.11.021.
Khan, A.W.; Lamb, K.A.; Schneider, H. Recovery of Fermentable Sugars from the Brewers’ Spent Grains by the Use of Fungal Enzymes. Process Biochem 1988, 23, 172–175.
Moreira, F.G.; Lima, F.A. de; Pedrinho, S.R.F.; Lenartovicz, V.; Souza, C.G.M. de; Peralta, R.M. Production of Amyla-ses by Aspergillus tamarii. Rev. Microbiol. 1999, 30, 157–162, doi:10.1590/S0001-37141999000200014
Mussatto, S.I. Brewer’s Spent Grain: A Valuable Feedstock for Industrial Applications. J Sci Food Agric 2014, 94, 1264–1275, doi:10.1002/jsfa.6486.
____________________________________________________
Prof. Dr. Maria de Lourdes T. M. Polizeli
Full Professor
University of São Paulo
Faculty of Philosophy, Sciences and Letters of Ribeirão Preto
Department of Biology
Phone.: +55 16 3315-4680
www.ffclrp.usp.br

Round 2
Reviewer 1 Report
Comments and Suggestions for Authors
Accept in present form
Reviewer 3 Report
Comments and Suggestions for Authors
To manuscript “Enhancing Xylanase Production from Aspergillus tamarii Kita and Its Application in the Bioconversion of Agro-Industrial 3 Residues into Fermentable Sugars Using Factorial Design”:
This research studied the xylanase production from Aspergillus tamarii Kita and further explored its application of three residues for fermentable sugar production. The methodology of central composite design is sound, and the data collection procedures are well-described. The data presented were informative and good for the xylanase production. The results of this study help to understand the production of xylanase from Aspergillus tamarii Kita and its application in the bioconversion of agricultural residues. After careful revision, I think it can be published in Fermentation.